# NON-AUTOREGRESSIVE NEURAL MACHINE TRANSLATION

**Jiatao Gu**[†,*] **James Bradbury**[‡]**, Caiming Xiong**[‡]**, Victor O.K. Li**[†] **& Richard Socher**[‡]

[‡]Salesforce Research
{james.bradbury,cxiong,rsocher}@salesforce.com
[†]The University of Hong Kong
{jiataogu, vli}@eee.hku.hk

## ABSTRACT

Existing approaches to neural machine translation condition each output word on previously generated outputs. We introduce a model that avoids this autoregressive property and produces its outputs in parallel, allowing an order of magnitude lower latency during inference. Through knowledge distillation, the use of input token fertilities as a latent variable, and policy gradient fine-tuning, we achieve this at a cost of as little as 2.0 BLEU points relative to the autoregressive Transformer network used as a teacher. We demonstrate substantial cumulative improvements associated with each of the three aspects of our training strategy, and validate our approach on IWSLT 2016 English–German and two WMT language pairs. By sampling fertilities in parallel at inference time, our non-autoregressive model achieves near-state-of-the-art performance of 29.8 BLEU on WMT 2016 English–Romanian.

## 1 INTRODUCTION

Neural network based models outperform traditional statistical models for machine translation (MT) (Bahdanau et al., 2015; Luong et al., 2015). However, state-of-the-art neural models are much slower than statistical MT approaches at inference time (Wu et al., 2016). Both model families use *autoregressive* decoders that operate one step at a time: they generate each token conditioned on the sequence of tokens previously generated. This process is not parallelizable, and, in the case of neural MT models, it is particularly slow because a computationally intensive neural network is used to generate each token.

While several recently proposed models avoid recurrence at train time by leveraging convolutions (Kalchbrenner et al., 2016; Gehring et al., 2017; Kaiser et al., 2017) or self-attention (Vaswani et al., 2017) as more-parallelizable alternatives to recurrent neural networks (RNNs), use of autoregressive decoding makes it impossible to take full advantage of parallelism during inference.

We introduce a non-autoregressive translation model based on the Transformer network (Vaswani et al., 2017). We modify the encoder of the original Transformer network by adding a module that predicts *fertilities*, sequences of numbers that form an important component of many traditional machine translation models (Brown et al., 1993). These fertilities are supervised during training and provide the decoder at inference time with a globally consistent plan on which to condition its simultaneously computed outputs.

## 2 BACKGROUND

### 2.1 AUTOREGRESSIVE NEURAL MACHINE TRANSLATION

Given a source sentence $X = \{x_1, ..., x_{T'}\}$, a neural machine translation model factors the distribution over possible output sentences $Y = \{y_1, ..., y_T\}$ into a chain of conditional probabilities with a

---

[*]This work was completed while the first author was interning at Salesforce Research.

left-to-right causal structure:

$$p_{\mathcal{AR}}(Y|X;\theta) = \prod_{t=1}^{T+1} p(y_t|y_{0:t-1}, x_{1:T'};\theta),\tag{1}$$

where the special tokens $y_0$ (e.g. $\langle\text{bos}\rangle$) and $y_{T+1}$ (e.g. $\langle\text{eos}\rangle$) are used to represent the beginning and end of all target sentences. These conditional probabilities are parameterized using a neural network. Typically, an encoder-decoder architecture (Sutskever et al., 2014) with a unidirectional RNN-based decoder is used to capture the causal structure of the output distribution.

**Maximum Likelihood training**    Choosing to factorize the machine translation output distribution autoregressively enables straightforward maximum likelihood training with a cross-entropy loss applied at each decoding step:

$$\mathcal{L}_{\text{ML}} = \log p_{\mathcal{AR}}(Y|X;\theta) = \sum_{t=1}^{T+1} \log p(y_t|y_{0:t-1}, x_{1:T'};\theta).\tag{2}$$

This loss provides direct supervision for each conditional probability prediction.

**Autoregressive NMT without RNNs**    Since the entire target translation is known at training time, the calculation of later conditional probabilities (and their corresponding losses) does not depend on the output words chosen during earlier decoding steps. Even though decoding must remain entirely sequential during inference, models can take advantage of this parallelism during training. One such approach replaces recurrent layers in the decoder with masked convolution layers (Kalchbrenner et al., 2016; Gehring et al., 2017) that provide the causal structure required by the autoregressive factorization.

A recently introduced option which reduces sequential computation still further is to construct the decoder layers out of self-attention computations that have been causally masked in an analogous way. The state-of-the-art Transformer network takes this approach, which allows information to flow in the decoder across arbitrarily long distances in a constant number of operations, asymptotically fewer than required by convolutional architectures (Vaswani et al., 2017).

## 2.2    Non-Autoregressive Decoding

**Pros and cons of autoregressive decoding**    The autoregressive factorization used by conventional NMT models has several benefits. It corresponds to the word-by-word nature of human language production and effectively captures the distribution of real translations. Autoregressive models achieve state-of-the-art performance on large-scale corpora and are easy to train, while beam search provides an effective local search method for finding approximately-optimal output translations.

But there are also drawbacks. As the individual steps of the decoder must be run sequentially rather than in parallel, autoregressive decoding prevents architectures like the Transformer from fully realizing their train-time performance advantage during inference. Meanwhile, beam search suffers from diminishing returns with respect to beam size (Koehn & Knowles, 2017) and exhibits limited search parallelism because it introduces computational dependence between beams.

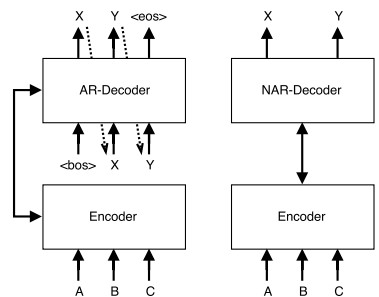

**Towards non-autoregressive decoding**    A naïve solution is to remove the autoregressive connection directly from an existing encoder-decoder model. Assuming that the target sequence length $T$ can be modeled with a separate conditional distribution $p_L$, this becomes

Figure 1:    Translating "A B C" to "X Y" using autoregressive and non-autoregressive neural MT architectures. The latter generates all output tokens in parallel.

$$p_{\mathcal{NA}}(Y|X;\theta) = p_L(T|x_{1:T'};\theta) \cdot \prod_{t=1}^{T} p(y_t|x_{1:T'};\theta).\tag{3}$$

This model still has an explicit likelihood function, and it can still be trained using independent cross-entropy losses on each output distribution. Now, however, these distributions can be computed in parallel at inference time.

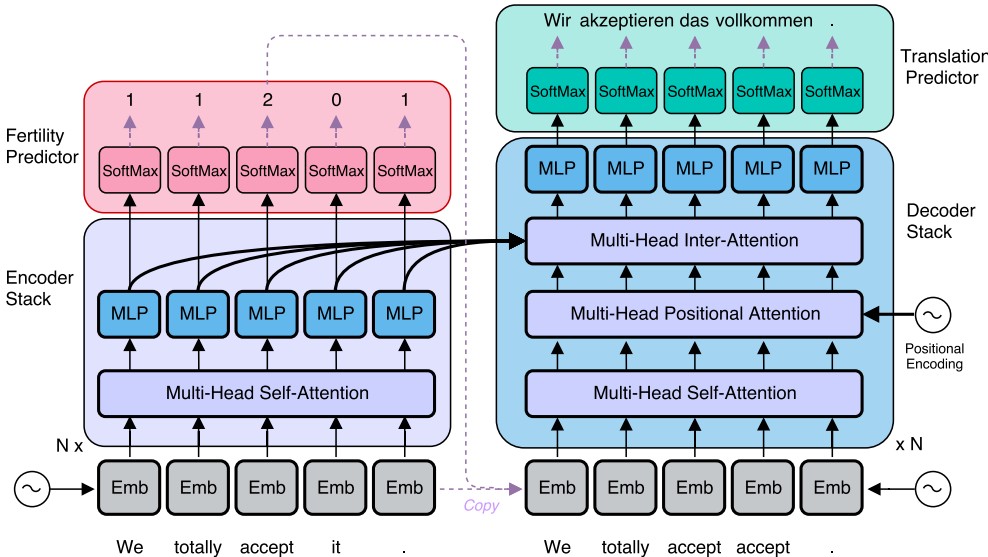

Figure 2: The architecture of the NAT, where the black solid arrows represent differentiable connections and the purple dashed arrows are non-differentiable operations. Each sublayer inside the encoder and decoder stacks also includes layer normalization and a residual connection.

## 2.3 THE MULTIMODALITY PROBLEM

However, this naïve approach does not yield good results, because such a model exhibits complete *conditional independence*. Each token's distribution $p(y_t)$ depends only on the source sentence $X$. This makes it a poor approximation to the true target distribution, which exhibits strong correlation across time. Intuitively, such a decoder is akin to a panel of human translators each asked to provide a single word of a translation independently of the words their colleagues choose.

In particular, consider an English source sentence like "Thank you." This can be accurately translated into German as any one of "Danke.", "Danke schön.", or "Vielen Dank.", all of which may occur in a given training corpus. This target distribution cannot be represented as a product of independent probability distributions for each of the first, second, and third words, because a conditionally independent distribution cannot allow "Danke schön." and "Vielen Dank." without also licensing "Danke Dank." and "Vielen schön."

The conditional independence assumption prevents a model from properly capturing the highly multimodal distribution of target translations. We call this the "multimodality problem" and introduce both a modified model and new training techniques to tackle this issue.

## 3 THE NON-AUTOREGRESSIVE TRANSFORMER (NAT)

We introduce a novel NMT model—the Non-Autoregressive Transformer (NAT)—that can produce an entire output translation in parallel. As shown in Fig. 2, the model is composed of the following four modules: an encoder stack, a decoder stack, a newly added fertility predictor (details in 3.3), and a translation predictor for token decoding.

### 3.1 ENCODER STACK

Similar to the autoregressive Transformer, both the encoder and decoder stacks are composed entirely of feed-forward networks (MLPs) and multi-head attention modules. Since no RNNs are used, there is no inherent requirement for sequential execution, making non-autoregressive decoding possible. For our proposed NAT, the encoder stays unchanged from the original Transformer network.

## 3.2 DECODER STACK

In order to translate non-autoregressively and parallelize the decoding process, we modify the decoder stack as follows.

**Decoder Inputs**   Before decoding starts, the NAT needs to know how long the target sentence will be in order to generate all words in parallel. More crucially, we cannot use time-shifted target outputs (during training) or previously predicted outputs (during inference) as the inputs to the first decoder layer. Omitting inputs to the first decoder layer entirely, or using only positional embeddings, resulted in very poor performance. Instead, we initialize the decoding process using copied source inputs from the encoder side. As the source and target sentences are often of different lengths, we propose two methods:

- **Copy source inputs uniformly**: Each decoder input $t$ is a copy of the Round$(T't/T)$-th encoder input. This is equivalent to "scanning" source inputs from left to right with a constant "speed," and results in a decoding process that is deterministic given a (predicted) target length.
- **Copy source inputs using fertilities**: A more powerful way, depicted in Fig. 2 and discussed in more detail below, is to copy each encoder input as a decoder input zero or more times, with the number of times each input is copied referred to as that input word's "fertility." In this case the source inputs are scanned from left to right at a "speed" that varies inversely with the fertility of each input; the decoding process is now conditioned on the sequence of fertilities, while the resulting output length is determined by the sum of all fertility values.

**Non-causal self-attention**   Without the constraint of an autoregressive factorization of the output distribution, we no longer need to prevent earlier decoding steps from accessing information from later steps. Thus we can avoid the causal mask used in the self-attention module of the conventional Transformer's decoder. Instead, we mask out each query position only from attending to itself, which we found to improve decoder performance relative to unmasked self-attention.

**Positional attention**   We also include an additional positional attention module in each decoder layer, which is a multi-head attention module with the same general attention mechanism used in other parts of the Transformer network, i.e.

$$\text{Attention}(Q, K, V) = \text{softmax}\left(\frac{QK^T}{\sqrt{d_{\text{model}}}}\right) \cdot V, \tag{4}$$

where $d_{\text{model}}$ is the model hidden size, but with the positional encoding[1] as both query and key and the decoder states as the value. This incorporates positional information directly into the attention process and provides a stronger positional signal than the embedding layer alone. We also hypothesize that this additional information improves the decoder's ability to perform local reordering.

## 3.3 MODELING FERTILITY TO TACKLE THE MULTIMODALITY PROBLEM

The multimodality problem can be attacked by introducing a latent variable $z$ to directly model the nondeterminism in the translation process: we first sample $z$ from a prior distribution and then condition on $z$ to non-autoregressively generate a translation.

One way to interpret this latent variable is as a sentence-level "plan" akin to those discussed in the language production literature (Martin et al., 2010). There are several desirable properties for this latent variable:

- It should be simple to infer a value for the latent variable given a particular input-output pair, as this is needed to train the model end-to-end.
- Adding $z$ to the conditioning context should account as much as possible for the correlations across time between different outputs, so that the remaining marginal probabilities at each output location are as close as possible to satisfying conditional independence.
- It should not account for the variation in output translations so directly that $p(y|x, z)$ becomes trivial to learn, since that is the function our decoder neural network will approximate.

---

[1]The positional encoding $p$ is computed as $p(j, k) = \sin\left(j/10000^{k/d}\right)$ (for even $k$) or $\cos\left(j/10000^{k/d}\right)$ (for odd $k$), where $j$ is the timestep index and $k$ is the channel index.

The factorization by length introduced in Eq. 3 provides a very weak example of a latent variable model, satisfying the first and third property but not the first. We propose the use of *fertilities* instead. These are integers for each word in the source sentence that correspond to the number of words in the target sentence that can be aligned to that source word using a hard alignment algorithm like IBM Model 2 (Brown et al., 1993).

One of the most important properties of the proposed NAT is that it naturally introduces an informative latent variable when we choose to copy the encoder inputs based on predicted fertilities. More precisely, given a source sentence $X$, the conditional probability of a target translation $Y$ is:

$$p_{\mathcal{NA}}(Y|X;\theta) = \sum_{f_1,...,f_{T'} \in \mathcal{F}} \left( \prod_{t'=1}^{T'} p_F(f_{t'}|x_{1:T'};\theta) \cdot \prod_{t=1}^{T} p(y_t|x_1\{f_1\},..,x_{T'}\{f_{T'}\};\theta) \right) \quad (5)$$

where $\mathcal{F} = \{f_1, ..., f_{T'}| \sum_{t'=1}^{T'} f_{t'} = T, f_{t'} \in \mathbb{Z}^*\}$ is the set of all fertility sequences—one fertility value per source word—that sum to the length of $Y$ and $x\{f\}$ denotes the token $x$ repeated $f$ times.

**Fertility prediction** As shown in Fig. 2, we model the fertility $p_F(f_{t'}|x_{1:T'})$ at each position independently using a one-layer neural network with a softmax classifier ($L = 50$ in our experiments) on top of the output of the last encoder layer. This models the way that fertility values are a property of each input word but depend on information and context from the entire sentence.

**Benefits of fertility** Fertilities possess all three of the properties listed earlier as desired of a latent variable for non-autoregressive machine translation:

- An external aligner provides a simple and fast approximate inference model that effectively reduces the unsupervised training problem to two supervised ones.

- Using fertilities as a latent variable makes significant progress towards solving the multimodality problem by providing a natural factorization of the output space. Given a source sentence, restricting the output distribution to those target sentences consistent with a particular fertility sequence dramatically reduces the mode space. Furthermore, the global choice of mode is factored into a set of local mode choices: namely, how to translate each input word. These local mode choices can be effectively supervised because the fertilities provide a fixed "scaffold."

- Including both fertilities and reordering in the latent variable would provide complete alignment statistics. This would make the decoding function trivially easy to approximate given the latent variable and force all of the modeling complexity into the encoder. Using fertilities alone allows the decoder to take some of this burden off of the encoder.

Our use of fertilities as a latent variable also means that there is no need to have a separate means of explicitly modeling the length of the translation, which is simply the sum of fertilities. And fertilities provide a powerful way to condition the decoding process, allowing the model to generate diverse translations by sampling over the fertility space.

### 3.4 TRANSLATION PREDICTOR AND THE DECODING PROCESS

At inference time, the model can identify the translation with the highest conditional probability (see Eq. 5) by marginalizing over all possible latent fertility sequences. Given a fertility sequence, however, identifying the optimal translation only requires independently maximizing the local probability for each output position. We define $Y = G(x_{1:T'}, f_{1:T'}; \theta)$ to represent the optimal translation given a source sentence and a sequence of fertility values.

But searching and marginalizing over the whole fertility space is still intractable. We propose three heuristic decoding algorithms to reduce the search space of the NAT model:

**Argmax decoding** Since the fertility sequence is also modeled with a conditionally independent factorization, we can simply estimate the best translation by choosing the highest-probability fertility for each input word:

$$\hat{Y}_{\text{argmax}} = G(x_{1:T'}, \hat{f}_{1:T'}; \theta), \text{where } \hat{f}_{t'} = \underset{f}{\text{argmax}}\, p_F(f_{t'}|x_{1:T'}; \theta) \quad (6)$$

**Average decoding**  We can also estimate each fertility as the expectation of its corresponding softmax distribution:

$$\hat{Y}_{\text{average}} = G(x_{1:T'}, \hat{f}_{1:T'}; \theta), \text{where } \hat{f}_{t'} = \text{Round}\left(\sum_{f_{t'}=1}^{L} p_F(f_{t'}|x_{1:T'}; \theta)f_{t'}\right) \tag{7}$$

**Noisy parallel decoding (NPD)**  A more accurate approximation of the true optimum of the target distribution, inspired by Cho (2016), is to draw samples from the fertility space and compute the best translation for each fertility sequence. We can then use the autoregressive teacher to identify the best overall translation:

$$\hat{Y}_{\text{NPD}} = G(x_{1:T'}, \operatorname*{argmax}_{f_{t'}\sim p_F} p_{\mathcal{AR}}(G(x_{1:T'}, f_{1:T'}; \theta)|X; \theta); \theta) \tag{8}$$

Note that, when using an autoregressive model as a scoring function for a set of decoded translations, it can run as fast as it does at train time because it can be provided with all decoder inputs in parallel.

NPD is a stochastic search method, and it also increases the computational resources required linearly by the sample size. However, because all the search samples can be computed and scored entirely independently, the process only doubles the latency compared to computing a single translation if sufficient parallelism is available.

## 4 TRAINING

The proposed NAT contains a discrete sequential latent variable $f_{1:T'}$, whose conditional posterior distribution $p(f_{1:T'}|x_{1:T'}, y_{1:T}; \theta)$ we can approximate using a proposal distribution $q(f_{1:T'}|x_{1:T'}, y_{1:T})$. This provides a variational bound for the overall maximum likelihood loss:

$$\mathcal{L}_{\text{ML}} = \log p_{\mathcal{NA}}(Y|X; \theta) = \log \sum_{f_{1:T'}\in\mathcal{F}} p_F(f_{1:T'}|x_{1:T'}; \theta) \cdot p(y_{1:T}|x_{1:T'}, f_{1:T'}; \theta)$$

$$\geq \operatorname*{\mathbb{E}}_{f_{1:T'}\sim q}\left(\underbrace{\sum_{t=1}^{T}\log p(y_t|x_1\{f_1\}, .., x_{T'}\{f_{T'}\}; \theta)}_{\text{Translation Loss}} + \underbrace{\sum_{t'=1}^{T'}\log p_F(f_{t'}|x_{1:T'}; \theta)}_{\text{Fertility Loss}}\right) + \mathcal{H}(q) \tag{9}$$

We choose a proposal distribution $q$ defined by a separate, fixed fertility model. Possible options include the output of an external aligner, which produces a deterministic sequence of integer fertilities for each (source, target) pair in a training corpus, or fertilities computed from the attention weights used in our fixed autoregressive teacher model. This simplifies the inference process considerably, as the expectation over $q$ is deterministic.

The resulting loss function, consisting of the two bracketed terms in Eq. 9, allows us to train the entire model in a supervised fashion, using the inferred fertilities to simultaneously train the translation model $p$ and supervise the fertility neural network model $p_F$.

### 4.1 SEQUENCE-LEVEL KNOWLEDGE DISTILLATION

While the latent fertility model substantially improves the ability of the non-autoregressive output distribution to approximate the multimodal target distribution, it does not completely solve the problem of nondeterminism in the training data. In many cases, there are multiple correct translations consistent with a single sequence of fertilities—for instance, both "Danke schön." and "Vielen dank." are consistent with the English input "Thank you." and the fertility sequence $[2, 0, 1]$, because "you" is not directly translated in either German sentence.

Thus we additionally apply sequence-level knowledge distillation (Kim & Rush, 2016) to construct a new corpus by training an autoregressive machine translation model, known as the teacher, on an

existing training corpus, then using that model's greedy outputs as the targets for training the non-autoregressive student. The resulting targets are less noisy and more deterministic, as the trained model will consistently translate a sentence like "Thank you." into the same German translation every time; on the other hand, they are also lower in quality than the original dataset.

## 4.2 FINE-TUNING

Our supervised fertility model enables a decomposition of the overall maximum likelihood loss into translation and fertility terms, but it has some drawbacks compared to variational training. In particular, it heavily relies on the deterministic, approximate inference model provided by the external alignment system, while it would be desirable to train the entire model, including the fertility predictor, end to end.

Thus we propose a fine-tuning step after training the NAT to convergence. We introduce an additional loss term consisting of the reverse K-L divergence with the teacher output distribution, a form of word-level knowledge distillation:

$$\mathcal{L}_{\text{RKL}}\left(f_{1:T'};\theta\right) = \sum_{t=1}^{T}\sum_{y_t}\left[\log p_{\mathcal{AR}}\left(y_t|\hat{y}_{1:t-1}, x_{1:T'}\right)\cdot p_{\mathcal{NA}}\left(y_t|x_{1:T'}, f_{1:T'};\theta\right)\right], \qquad (10)$$

where $\hat{y}_{1:T} = G(x_{1:T'}, f_{1:T'};\theta)$. Such a loss is more favorable towards highly peaked student output distributions than a standard cross-entropy error would be.

Then we train the whole model jointly with a weighted sum of the original distillation loss and two such terms, one an expectation over the predicted fertility distribution, normalized with a baseline, and the other based on the external fertility inference model:

$$\mathcal{L}_{\text{FT}} = \lambda\left(\underbrace{\mathbb{E}_{f_{1:T'}\sim p_F}\left(\mathcal{L}_{\text{RKL}}\left(f_{1:T'}\right) - \mathcal{L}_{\text{RKL}}\left(\bar{f}_{1:T'}\right)\right)}_{\mathcal{L}_{\text{RL}}} + \underbrace{\mathbb{E}_{f_{1:T'}\sim q}\left(\mathcal{L}_{\text{RKL}}\left(f_{1:T'}\right)\right)}_{\mathcal{L}_{\text{BP}}}\right) + (1-\lambda)\mathcal{L}_{\text{KD}} \quad (11)$$

where $\bar{f}_{1:T'}$ is the average fertility computed by Eq. 7. The gradient with respect to the non-differentiable $\mathcal{L}_{\text{RL}}$ term can be estimated with REINFORCE (Williams, 1992), while the $\mathcal{L}_{\text{BP}}$ term can be trained using ordinary backpropagation.

# 5 EXPERIMENTS

## 5.1 EXPERIMENTAL SETTINGS

**Dataset** We evaluate the proposed NAT on three widely used public machine translation corpora: IWSLT16 En–De[2], WMT14 En–De,[3] and WMT16 En–Ro[4]. We use IWSLT—which is smaller than the other two datasets—as the development dataset for ablation experiments, and additionally train and test our primary models on both directions of both WMT datasets. All the data are tokenized and segmented into subword symbols using byte-pair encoding (BPE) (Sennrich et al., 2015) to restrict the size of the vocabulary. For both WMT datasets, we use shared BPE vocabulary and additionally share encoder and decoder word embeddings; for IWSLT, we use separate English and German vocabulary and embeddings.

**Teacher** Sequence-level knowledge distillation is applied to alleviate multimodality in the training dataset, using autoregressive models as the teachers. The same teacher model used for distillation is also used as a scoring function for fine-tuning and noisy parallel decoding.

To enable a fair comparison, and benefit from its high translation quality, we implemented the autoregressive teachers using the state-of-the-art Transformer architecture. In addition, we use the same sizes and hyperparameters for each student and its respective teacher, with the exception of the newly added positional self-attention and fertility prediction modules.

---

[2]https://wit3.fbk.eu/
[3]http://www.statmt.org/wmt14/translation-task
[4]http://www.statmt.org/wmt16/translation-task

| Models | WMT14 | | WMT16 | | IWSLT16 | | |
|---|---|---|---|---|---|---|---|
| | En→De | De→En | En→Ro | Ro→En | En→De | Latency / Speedup | |
| NAT | 17.35 | 20.62 | 26.22 | 27.83 | 25.20 | 39 ms | 15.6× |
| NAT (+FT) | 17.69 | 21.47 | 27.29 | 29.06 | 26.52 | 39 ms | 15.6× |
| NAT (+FT + NPD $s = 10$) | 18.66 | 22.41 | 29.02 | 30.76 | 27.44 | 79 ms | 7.68× |
| NAT (+FT + NPD $s = 100$) | 19.17 | 23.20 | 29.79 | **31.44** | 28.16 | 257 ms | 2.36× |
| Autoregressive ($b = 1$) | 22.71 | 26.39 | 31.35 | 31.03 | 28.89 | 408 ms | 1.49× |
| Autoregressive ($b = 4$) | 23.45 | 27.02 | 31.91 | 31.76 | 29.70 | 607 ms | 1.00× |

Table 1: BLEU scores on official test sets (`newstest2014` for WMT En-De and `newstest2016` for WMT En-Ro) or the development set for IWSLT. NAT models without NPD use argmax decoding. Latency is computed as the time to decode a single sentence without minibatching, averaged over the whole test set; decoding is implemented in PyTorch on a single NVIDIA Tesla P100.

**Preparation for knowledge distillation**   We first train all teacher models using maximum likelihood, then freeze their parameters. To avoid the redundancy of running fixed teacher models repeatedly on the same data, we decode the entire training set once using each teacher to create a new training dataset for its respective student.

**Encoder initialization**   We find it helpful to initialize the weights in the NAT student's encoder with the encoder weights from its teacher, as the autoregressive and non-autoregressive models share the same encoder input and architecture.

**Fertility supervision during training**   As described above, we supervise the fertility predictions at train time by using a fixed aligner as a fertility inference function. We use the `fast_align`[5] implementation of IBM Model 2 for this purpose, with default parameters (Dyer et al., 2013).

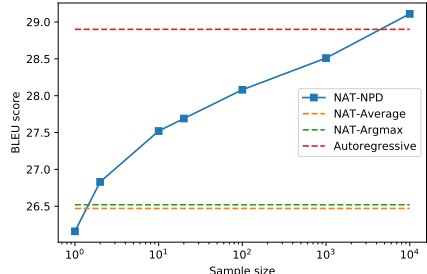

Figure 3:   BLEU scores on IWSLT development set as a function of sample size for noisy parallel decoding. NPD matches the performance of the other two decoding strategies after two samples, and exceeds the performance of the autoregressive teacher with around 1000.

**Hyperparameters**   For experiments on WMT datasets, we use the hyperparameter settings of the `base` Transformer model described in Vaswani et al. (2017), though without label smoothing. As IWSLT is a smaller corpus, and to reduce training time, we use a set of smaller hyperparameters ($d_{\text{model}} = 287, d_{\text{hidden}} = 507, n_{\text{layer}} = 5, n_{\text{head}} = 2$, and $t_{\text{warmup}} = 746$) for all experiments on that dataset. For fine-tuning we use $\lambda = 0.25$.

**Evaluation metrics**   We evaluate using tokenized and cased BLEU scores (Papineni et al., 2002).

**Implementation**   We have open-sourced our PyTorch implementation of the NAT[6].

## 5.2   RESULTS

Across the three datasets we used, the NAT performs between 2-5 BLEU points worse than its autoregressive teacher, with part or all of this gap addressed by the use of noisy parallel decoding. In the case of WMT16 English–Romanian, NPD improves the performance of our non-autoregressive model to within 0.2 BLEU points of the previous overall state of the art (Gehring et al., 2017).

Comparing latencies on the development model shows a speedup of more than a factor of 10 over greedy autoregressive decoding, or a factor of 15 over beam search. Latencies for decoding with NPD, regardless of sample size, could be reduced to about 80ms by parallelizing across multiple GPUs because each sample can be generated, then scored, independently from the others.

---

[5]https://github.com/clab/fast_align
[6]https://github.com/salesforce/nonauto-nmt

## 5.3 Ablation Study

We also conduct an extensive ablation study with the proposed NAT on the IWSLT dataset. First, we note that the model fails to train when provided with only positional embeddings as input to the decoder. Second, we see that training on the distillation corpus rather than the ground truth provides a fairly consistent improvement of around 5 BLEU points. Third, switching from uniform copying of source inputs to fertility-based copying improves performance by four BLEU points when using ground-truth training or two when using distillation.

| Distillation | | Decoder Inputs | | +PosAtt | Fine-tuning | | | BLEU | BLEU (T) |
| b=1 | b=4 | +uniform | +fertility | | $+\mathcal{L}_{\mathrm{KD}}$ | $+\mathcal{L}_{\mathrm{BP}}$ | $+\mathcal{L}_{\mathrm{RL}}$ | | |
|---|---|---|---|---|---|---|---|---|---|
| | | | | ✓ | | | | $\approx 2$ | |
| | | ✓ | | ✓ | | | | 16.51 | |
| | | | ✓ | ✓ | | | | 18.87 | |
| ✓ | | ✓ | | ✓ | | | | 20.72 | |
| | ✓ | ✓ | | ✓ | | | | 21.12 | |
| ✓ | | | ✓ | | | | | 24.02 | 43.91 |
| ✓ | | | ✓ | ✓ | | | | 25.20 | 45.41 |
| ✓ | | ✓ | | ✓ | ✓ | ✓ | | 22.44 | |
| ✓ | | | ✓ | ✓ | | | ✓ | × | × |
| ✓ | | | ✓ | ✓ | | ✓ | | × | × |
| ✓ | | | ✓ | ✓ | ✓ | ✓ | | 25.76 | 46.11 |
| ✓ | | | ✓ | ✓ | ✓ | ✓ | ✓ | **26.52** | **47.38** |

Table 2: Ablation performance on the IWSLT development set. BLEU (T) refers to the BLEU score on a version of the development set that has been translated by the teacher model. An × indicates that fine-tuning caused that model to get worse. When uniform copying is used as the decoder inputs, the ground-truth target lengths are provided. All models use argmax decoding.

Fine-tuning does not converge with reinforcement learning alone, or with the $\mathcal{L}_{\mathrm{BP}}$ term alone, but use of all three fine-tuning terms together leads to an improvement of around 1.5 BLEU points. Training the student model from a distillation corpus produced using beam search is similar to training from the greedily-distilled corpus.

| Source: | politicians try to pick words and use words to shape reality and control reality , but in fact , reality changes words far more than words can ever change reality . |
|---|---|
| Target: | Politiker versuchen Worte zu benutzen , um die Realität zu formen und die Realität zu kontrollieren , aber tatsächlich verändert die Realität Worte viel mehr , als Worte die Realität jemals verändern könnten . |
| AR: | Politiker versuchen Wörter zu wählen und Wörter zur Realität zu gestalten und Realität zu steuern , aber in Wirklichkeit verändert sich die Realität viel mehr als Worte , die die Realität verändern können . |
| NAT: | Politiker versuchen , Wörter wählen und zu verwenden , um Realität zu formen und Realität zu formen , aber tatsächlich ändert Realität Realität viel mehr als Worte die Realität Realität verändern . |
| NAT+NPD: | Politiker versuchen , Wörter wählen und zu verwenden , um Realität Realität formen und die Realität zu formen , aber tatsächlich ändert die Realität Worte viel mehr als Worte jemals die Realität verändern können . |
| Source: | I see wheelchairs bought and sold like used cars . |
| Target: | ich erlebe , dass Rollstühle gekauft und verkauft werden wie Gebrauchtwagen |
| AR: | ich sehe Rollstühlen , die wie Autos verkauft und verkauft werden . |
| NAT: | ich sehe , dass Stühle Stühle und verkauft wie Autos verkauft . |
| NAT+NPD: | ich sehe Rollühle kauften und verkaufte wie Autos . |

Figure 4: Two examples comparing translations produced by an autoregressive (AR) and non-autoregressive Transformer as well as the result of noisy parallel decoding with sample size 100. Repeated words are highlighted in gray.

We include two examples of translations from the IWSLT development set in Fig. 4. Instances of repeated words or phrases, highlighted in gray, are most prevalent in the non-autoregressive output for the relatively complex first example sentence. Two pairs of repeated words in the first example, as

| se lucreaza la solutii de genul acesta . | |
|---|---|
| se la solutii de genul acesta . | solutions on this kind are done . |
| se lucreaza la solutii de acesta . | work done on solutions like this . |
| se lucreaza solutii de genul acesta . | solutions on this kind is done . |
| se se lucreaza la solutii de acesta . | work is done on solutions like this . |
| se lucreaza lucreaza la solutii de acesta . | work is done on solutions like this . |
| se se lucreaza lucreaza la solutii de acesta . | **work is being done on solutions like this .** |
| se se lucreaza lucreaza la solutii de de acesta . | work is being done on solutions such as this . |
| se se lucreaza lucreaza la solutii de genul acesta . | work is being done on solutions such this kind . |

Figure 5: A Romanian–English example translated with noisy parallel decoding. At left are eight sampled fertility sequences from the encoder, represented with their corresponding decoder input sequences. Each of these values for the latent variable leads to a different possible output translation, shown at right. The autoregressive Transformer then picks the best translation, shown in red, a process which is much faster than directly using it to generate output.

well as a pair in the second, are not present in the versions with noisy parallel decoding, suggesting that NPD scoring using the teacher model can filter out such mistakes. The translations produced by the NAT with NPD, while of a similar quality to those produced by the autoregressive model, are also noticeably more literal.

We also show an example of the noisy parallel decoding process in Fig. 5, demonstrating the diversity of translations that can be found by sampling from the fertility space.

## 6 CONCLUSION

We introduce a latent variable model for non-autoregressive machine translation that enables a decoder based on Vaswani et al. (2017) to take full advantage of its exceptional degree of internal parallelism even at inference time. As a result, we measure translation latencies of one-tenth that of an equal-sized autoregressive model, while maintaining competitive BLEU scores.

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

## A    SCHEMATIC AND ANALYSIS

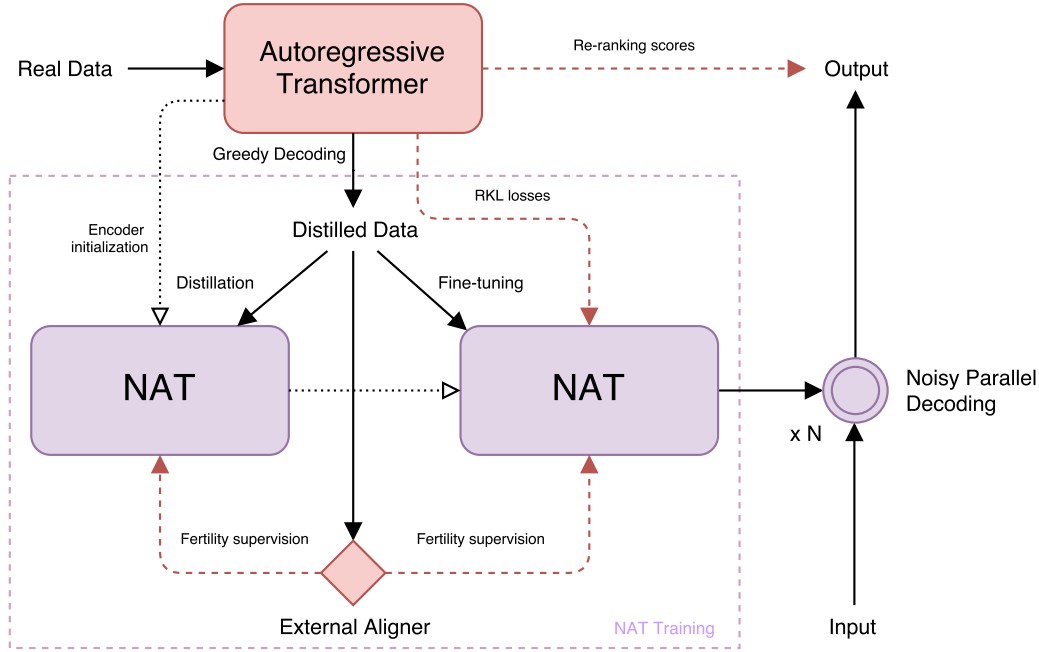

Figure 6: The schematic structure of training and inference for the NAT. The "distilled data" contains target sentences decoded by the autoregressive model and ground-truth source sentences.

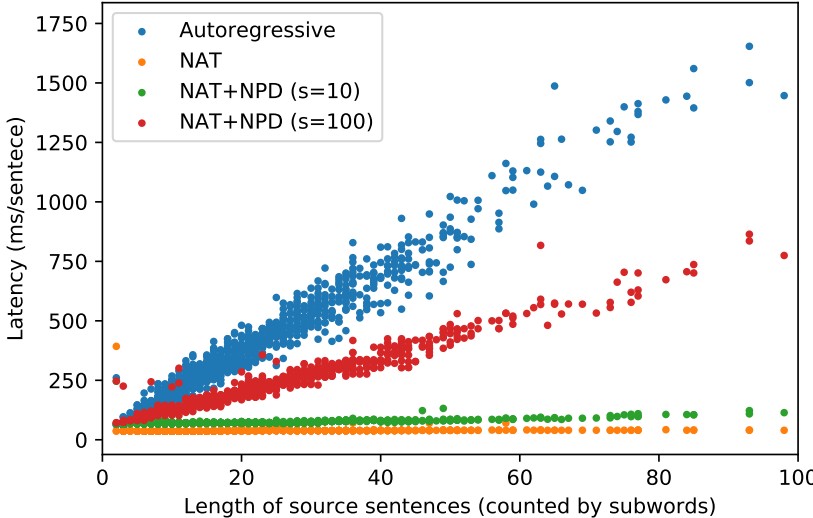

Figure 7: The translation latency, computed as the time to decode a single sentence without mini-batching, for each sentence in the IWSLT development set as a function of its length. The autoregressive model has latency linear in the decoding length, while the latency of the NAT is nearly constant for typical lengths, even with NPD with sample size 10. When using NPD with sample size 100, the level of parallelism is enough to more than saturate the GPU, leading again to linear latencies.

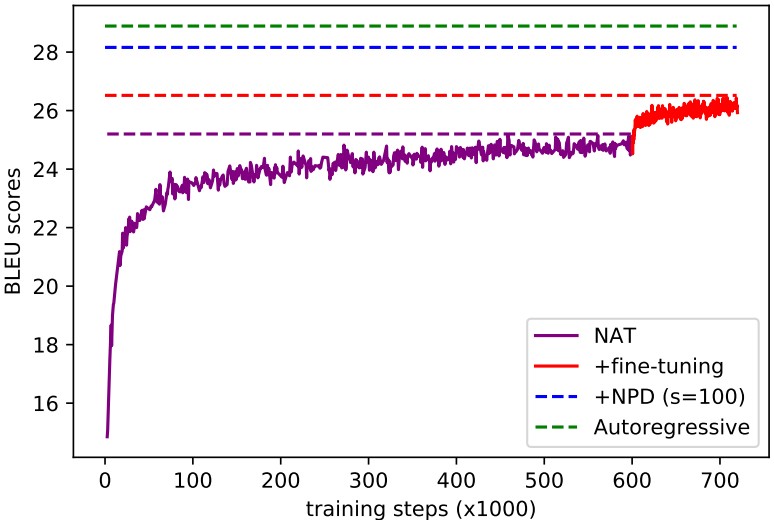

Figure 8: Learning curves for training and fine-tuning of the NAT on IWSLT. BLEU scores are on the development set.

