# OpenReview forum: "Non-Autoregressive Neural Machine Translation"
_ICLR.cc/2018/Conference — Accept (Poster)_

### Official Review · AnonReviewer3 · 2017-11-27
**Interesting work**

**Rating:** 7
**Confidence:** 4

**Review:**

This work proposes non-autoregressive decoder for the encoder-decoder framework in which the decision of generating a word does not depends on the prior decision of generated words. The key idea is to model the fertility of each word so that copies for source words are fed as input to the encoder part, not the generated target words as inputs. To achieve the goal, authors investigated various techniques: For inference, sample fertility space for generating multiple possible translations. For training, apply knowledge distilation for better training followed by fine tuning by reinforce. Experiments for English/German and English/Romanian show comparable translation qualities with speedup by non-autoregressive decoding.

The motivation is clear and proposed methods are very sound. Experiments are carried out very carefully.

I have only minor concerns to this paper:

- The experiments are designed to achieve comparable BLEU with improved latency. I'd like to know whether any BLUE improvement might be possible under similar latency, for instance, by increasing the model size given that inference is already  fast enough.

- It'd also like to see other language pairs with distorted word alignment, e.g., Chinese/English, to further strengthen this work, though  it might have little impact given that attention already capture sort of alignment.

- What is the impact of the external word aligner quality? For instance, it would be possible to introduce a noise in the word alignment results or use smaller data to train a model for word aligner.

- The positional attention is rather unclear and it would be better to revise it. Note that equation 4 is simply mentioning attention computation, not the proposed positional attention.

---

> ### Author Response · Authors · 2018-01-03
> **Thoughtful review and interesting questions**
>
> As explained in the response to Reviewer 2, we decided to standardize on a single model size for the WMT experiments, but acknowledge that an evaluation of comparative performance at different sizes would be a worthwhile follow-up. However, direct comparison of the NAT at one model size to an autoregressive Transformer at a different model size may not be especially informative, because our NAT relies on an autoregressive teacher with the same model size in order to initialize the encoder.
>
> The presence of the model distillation step also suggests that a meaningful BLEU improvement from larger NAT model size is unlikely, since only the NPD step allows the NAT to outperform its autoregressive teacher, even in the best case.
>
> We believe that the NAT's gap in performance between the English/German language pair and the English/Romanian language pair suggests that it is sensitive to the degree of reordering; we agree that it would be worthwhile to follow up on this hypothesis with pairs like Japanese/English or Turkish/English that exhibit even more reordering than German/English does.
>
> The external aligner we used produces fairly noisy results; our experiments with using the attention weights from an autoregressive Transformer as a (potentially more powerful) alignment model resulted in somewhat worse performance, suggesting that the dependence on alignment quality may not be straightforward.
>
> We can revise our description of the positional attention layer.

---

### Official Review · AnonReviewer2 · 2017-11-27
**review of "Non-autoregressive neural machine translation"**

**Rating:** 7
**Confidence:** 4

**Review:**

This paper describes an approach to decode non-autoregressively for neural machine translation (and other tasks that can be solved via seq2seq models). The advantage is the possibility of more parallel decoding which can result in a significant speed-up (up to a factor of 16 in the experiments described). The disadvantage is that it is more complicated than a standard beam search as auto-regressive teacher models are needed for training and the results do not reach (yet) the same BLEU scores as standard beam search.

Overall, this is an interesting paper. It would have been good to see a speed-accuracy curve which plots decoding speed for different sized models versus the achieved BLUE score on one of the standard benchmarks (like WMT14 en-fr or en-de) to understand better the pros and cons of the proposed approach and to be able to compare models at the same speed or the same BLEU scores. Table 1 gives a hint of that but it is not clear whether much smaller models with standard beam search are possibly as good and fast as NAT -- losing 2-5 BLEU points on WMT14 is significant.  While the Ro->En results are good, this particular language pair has not been used much by others; it would have been more interesting to stay with a single well-used language pair and benchmark and analyze why WMT14 en->de and de->en are not improving more. Finally it would have been good to address total computation in the comparison as well -- it seems while total decoding time is smaller total computation for NAT + NPD is actually higher depending on the choice of s.

---

> ### Author Response · Authors · 2018-01-03
> **Thoughtful review with many excellent points**
>
> The reviewer has brought up an interesting point about comparison of models at different sizes.
>
> We agree that the gap on English/German WMT14 is large enough that a relatively smaller autoregressive Transformer, especially on short sentences and with highly optimized inference kernels, might achieve similar latency and accuracy to the NAT. But no amount of kernel optimization or model size reduction can change the sequential nature of autoregressive translation; the autoregressive latency will always be proportional to the sentence length and the NAT will be faster when sequences are sufficiently long. The non-autoregressive Transformer can also benefit from low-level optimizations like quantization; we believe we compared the two on an even footing by using similar implementations for both.
>
> Also, while the original Transformer paper provided a strong set of baseline hyperparameters for the autoregressive architecture given a particular model size, we would need to conduct a significant amount of additional search to identify the right parameter settings for other model sizes. Instead we chose to focus our computational resources on the ablation study and more language pairs.
>
> We think the difference between the EN<–>DE and EN<–>RO results may be the result of a greater need for long-distance (clause-level) reordering between English and German (which are closely related languages with significant differences in sentence structure) than between English and Romanian (which, while less closely related, have more similarities in word order); this is an interesting direction for future research.
>
> As for computation time, we are making the assumption that a significant amount of parallelism is available and the primary metric is the latency on the critical path. This is not necessarily the case for every deployment of machine translation in practice, but it is a metric on which existing neural MT systems perform particularly poorly. Given that assumption, the additional computation needed for NPD, while potentially significant in terms of throughput, would not result in more than a doubling of latency.

---

### Official Review · AnonReviewer1 · 2017-11-29
**Fast inference for transformer nmt**

**Rating:** 6
**Confidence:** 4

**Review:**

This paper can be seen as an extension of the paper "attention is all you need" that will be published at nips in a few weeks (at the time I write this review).

The goal here is to make the target sentence generation non auto regressive. The authors propose to introduce a set of latent variables to represent the fertility of each source words. The number of target words can be then derived and they're all predicted in parallel.

The idea is interesting and trendy. However, the paper is not really stand alone. A lot of tricks are stacked to reduce the performance degradation. However, they're sometimes to briefly described to be understood by most readers.

The training process looks highly elaborate with a lot of hyper parameters. Maybe you could comment on this.

For instance, the use fertility supervision during training could be better motivated and explained. Your choice of IBM 2 is wired since it doesn't include fertility. Why not IBM 4, for instance ? How you use IBM model for supervision. This a simple example, but a lot of things in this paper is too briefly described and their impact not really evaluated.

---

> ### Author Response · Authors · 2018-01-03
> **Reviewer raises good points**
>
> We agree that the methodology presented in this paper contains several moving parts and a few techniques, such as external alignment supervision, sequence-level knowledge distillation, and fine-tuning using reinforcement learning, that could be considered “tricks.” All of these techniques are targeted at solving the multi-modality problem introduced when performing non-autoregressive translation, but if there’s a particular part of the pipeline that you feel is unclear, we would be happy to improve the description and explanation.
>
> Also, we would argue that our approach introduces relatively few additional hyperparameters over the original Transformer, primarily the inclusion of the various fine-tuning losses and the number of fertility samples used at inference time. While we did not conduct an exhaustive search over e.g. a range of possible values for the weights on each fine-tuning loss, we tried to present a reasonably comprehensive set of ablations to identify the effect of each part of our methodology.
>
> We also agree that IBM 2 might not be the best possible choice of fertility inference model, since the model itself only considers fertility implicitly (as part of the alignment process) and not explicitly like IBM 3+. Our decision to use IBM 2 was based on the availability, performance, and ease of integration of a popular existing implementation (fast_align) of that particular alignment model. Meanwhile, the use of fertility supervision in the first place can be justified from two perspectives: from the variational inference perspective, an external alignment model provides a very simple and tractable proposal distribution; at a higher level, fertility supervision simply turns a difficult, unsupervised learning problem into an easier, supervised one.

---

### Public Comment · ~Ozan_Caglayan1 · 2017-11-21
**Two small glitches**

Page 3, Paragraph 1: "The factorization by length introduced .... first and third property but not the **first.**"
Page 6,  4.1.: Why is the fertility sequence [2,0,1] ? If I understood fertility correctly, I think it should have been [2,0] since number of tokens in the source sentence is 2.

---

> ### Author Response · Authors · 2017-11-21
> **Thanks!**
>
> You're right about the first error; that's a typo. For the second point, the fertility sequence is [2, 0, 1] because our analysis (and the network) counts the period/full stop as a third source/target token.

---

### Decision · Program_Chairs · 2018-01-29
**ICLR 2018 Conference Acceptance Decision**

**Decision:**

Accept (Poster)

**Comment:**

The paper proposes a novel method for training a non-autoregressive machine translation model based on a pre-trained auto-regressive model. The method is interesting and the evaluation is carried out well. It should be noted, however, that the relative complexity of the training procedure (which involves multiple stages and external supervision) might limit the practical applicability and impact of the technique.